# How the persistence of patriarchy undermines the financial empowerment of women microfinance borrowers? Evidence from a southern sub-district of Bangladesh

Tunvir Ahamed Shohel [1,2]*, Sara Niner[3], Samanthi Gunawardana[4]

**1** PhD Student at Monash University, Melbourne, Australia, **2** Faculty, Khulna University, Gollamari, Bangladesh, **3** Lecturer, School of Social Sciences, Monash University, Melbourne, Australia, **4** Senior Lecturer, School of Social Sciences, Director, Master of International Development Practice, Monash University, Melbourne, Australia

* tunvir.shohel@monash.edu, tunvirshohel@ku.ac.bd

## Abstract

A significant body of multi-disciplinary research supports the proposition that women may experience empowerment from microfinance programs. This is based on the assumption that an increase in women's financial contribution to the household helps to transform gender norms and relations which increases their decision-making power. However, the relationship between the strength and persistence of patriarchal gender norms within the household and women's financial empowerment needs further exploration. This paper presents the findings of a mixed-method study comprising 331 surveys and 33 in-depth interviews with women receiving microfinance and their husbands in a southern sub-district of Bangladesh; it draws upon gender socialisation and gender performance theory to understand how patriarchal gender norms influence women's financial empowerment in households receiving microfinance. Findings demonstrate that participation in microfinance programs has not shifted gender norms, nor financially empowered women. Women's loans were largely controlled by men as prescribed by underlying, unchanged patriarchal gender norms. The inter-generational reproduction of patriarchal gender relations continued to reproduce a strict gendered division of labour that reinforced restrictions on women's behaviour, mobility, and decision-making domains, and men's dominance in household and economic decision-making.

## Introduction

Globally, women disproportionately suffer from economic discrimination (e.g., restriction from financial dealings) and exploitation (e.g., unpaid domestic labour) [1–4]. Development scholars and practitioners from diverse disciplinary backgrounds identify the lack of women's access to financial resources as one of the major impediment to their empowerment [5,6].

Over the last few decades, developing countries' have sought to address this issue by implementing financial inclusion programs for women. The most ubiquitous of these have been microfinance programs. Microfinance refers to a series of financial services (e.g., small loans,

**Data Availability Statement:** Data for this paper has been made public and accessible via the doi link:- https://doi.org/10.7910/DVN/2MONFQ.

**Funding:** This article is part of a PhD research. This research is not funded by any external organisation(s).

**Competing interests:** There is no competing interest found.

insurance, savings; note: savings programs are also part of microfinance products in Bangladesh [7]) provided to those who traditionally lack collateral to access money from the formal banking system [8]. Microfinance has been widely recognised as an ideal type of aid and development intervention for enhancing women's financial empowerment by increasing women's income, financial contribution to household and financial decision-making power in the family [1,5,9–14]. The assumption is that women would use loans to begin, sustain or expand income-generating activities (IGAs) and gain financial empowerment for themselves and their household [12,13,15].

Studies on women's financial empowerment versus women's empowerment through microfinance programs have emerged in different fields. Among these, four bodies of scholarship document microfinance programs' emancipatory financial approach to predict and evaluate women's financial participation and empowerment. First, there are institutions such as the World Bank (WB), Non-government Organisations (NGOs), Microfinance Institutions (MFIs) and affiliated researchers such as Khandker [1], Christen, Rhyne [16] that have advocated for the microfinance model for women's empowerment under a trajectory of neoliberal economic growth. Second, economists such as Woller, Dunford [17], Morduch [18], Hashemi, Schuler [19] and Wilson [20] have quantitatively evaluated microfinance programs and found positive outcomes, including evidence for reduced rates of poverty. Third, multi-disciplinary development studies scholars highlight the positive impact of microfinance programs over several decades [e.g., M. Alam, 1988; Bangladesh Institute of Development Studies (BIDS), 1985, as cited in Bangoura ([21], Ferguson [22], Osmani [23], Chowdhury and Bhuiya [24], Moser [25], Jabbar, Shohel [26]). However, others e.g., [27–30] find that the woman-focused lending model fails to impact positively on its borrowers due to men's control over loan, insignificant income increase and increasing debt. Finally, several scholars from the social sciences (e.g., political economy, sociology, anthropology, human geography, social work and gender studies) advance critiques of the microfinance industry, citing evidence that socio-cultural factors such as patriarchy is significant to understand the impact of microfinance on women's financial empowerment outcomes [31–33].

In this paper, we define patriarchy as "a system of power in which male privilege and superiority over women are manifested, institutionalised, and self-reproducing across a society as a whole" Shepherd [34]. Lerner [35] as cited in Chowdhury [36] defines patriarchy as a historical creation which places the family as a core unit and basic foundation of social organisation by assembling gender roles for different sexes at households. Although the exact definition of patriarchy is contested, we find it a useful concept because power relations between genders are constructed at household level through (re)construction and (re)production of gender roles [35,37,38] and gender (re)performativity [37,39,40], particularly on the aspect of gender socialisation practices in the families. In other words, households are important sites of self-reproduction for patriarchy.

This mixed-method, multidisciplinary study examines the impact of gender norms on the financial empowerment of women recipients of microfinance in rural Bangladesh. We use gender socialisation and gender performance theory to understand the underlying gender norms that underscore everyday gendered practices in households. Gender socialisation theory argues that an individual's gender identity is formed within social structures and interactions [41]. Gender performance theory further elaborates gender socialisation theory by arguing that by performing gender roles in everyday interactions, long-lasting gender norms are enacted, produced, reproduced and sustained within social structures [37,39–41]. The process is so mundane that "we seldom question its presence in every facet of our daily lives". Relevant literature and our theoretical proposition add objective of this study to see how the microfinance borrowing women adjust financial dealings with the entranced patriarchy and its driven

gender norms and practices. Findings from this study would provide important basis for policy and development interventions for microfinance operations.

This paper is structured as follows. First, the literature concerning women's empowerment through microfinance programs is reviewed, with attention to the analytical differences between economists and other social scientists (e.g., sociology, anthropology, development studies). The theoretical framework on gender socialisation and performance within families in patriarchal societies is then presented and discussed, followed by the methodology and findings of the research study, discussion of findings and concluding remarks. Additionally, relevant sociological and development studies literature is reviewed to provide further analysis of the microfinance empowerment programs studied and if and how they engage patriarchal norms.

## Microfinance and women's empowerment

Different bodies of multi-disciplinary literature [27,42–44] critically investigates the empowerment outcomes of women's microfinance programs. However, the assessment of whether women were empowered by participating in microfinance depends on how scholars conceptualise 'women's empowerment', the methodologies used and the epistemological approach to interpreting findings.

Many institutions (e.g., IMF, World bank, MFIs) and affiliated scholars (e.g., Khandker [1], Christen, Rhyne [16], Malhotra [45], Barnes, Keogh [46]) conceptualise women's financial improvement as women's empowerment. According to this view, the primary obstacle to women's empowerment is inadequate access to financial resources. Since microfinance provides women with access to finance, women are understood to gain greater control over their economic state, with greater opportunities to escape poverty. In a similar vein, economist such as Hossain [47], Pitt and Khandker [43], Yunus and Jolis [48], Hashemi and Morshed [49], Morduch and Haley [50], Snodgrass and Sebstad [51], Alam [52], use quantitative measures such as: 1) an increase of borrower women's income; 2) investment with return and loan repayment rate; 3) the length of women's participation in microfinance schemes, and; 4) MFI's loan disbursement and recovery rate to assert the equivalence between women's financial gain and women's empowerment.

From a different lens, development studies scholars such as Osmani [23], Moser [25], Goetz and Gupta [27], Karnani [29] argue for the need to examine women's empowerment within a more holistic framework, such as human development (e.g., health, nutrition, gender power relations). For example, Osmani [23] finds that women's financial empowerment gained through microfinance participation over longer time periods provides women with greater human development gains such as freedom of movement and autonomy and in higher household contribution (e.g., spending money in nutrition, health), while Moser [25] also finds enhanced decision-making power in the household and Chowdhury and Bhuiya [24] further indicate positive outcomes for children's education.

However, other development scholars are critical, arguing that microfinance programs can have no or negative impacts on women's human development outcomes. This includes issues such as an insignificant rate of increased income [28,53], little or no impact on female enterprise expansion [29,54,55], men's control over women's loans [27], and more debt burdens for marginalised poor women [56].

Considering these negative outcomes from microfinance participation, an extant body of literature indicates that patriarchal constructs, such as gender relations and norms, socio-economic status and marriage customs are crucial to understanding the outcomes of microfinance for women [57]. Patriarchal expectations and norms about women's role in the household and

wider society appear particularly important. For example, sociological literature suggests that cultural norms about women's role in financial management may mean the female borrower passes on the loan funds for management and control to male relatives [31]. In this regard, Isserles [31] validates Goetz and Gupta [27] study finding from Bangladesh, which found that male family members had control over microfinance loans. Men were found to be utilizing loans making women 'vessels for men's economic activity' [27] Moreover, women losing control of loans not only reinforced the prevailing discriminatory gendered structures but became an added burden for women instead of empowering them [31]. An anthropological study from Timor-Leste by Niner [33] further indicates that gendered expectations about women's domestic role often led them to spend their loans on household expenses, while men invest more in large-scale business. Rahman [58] in a study focused on Bangladesh, found that limitations on women's freedom of movement allows MFIs to easily capture or communicate with women for loan recovery purpose. Karim [32,59] showed that in Bangladesh, loans were used by the men, although women were formally liable to repay the money. MFIs could easily apply strategies such as public shaming, threatening women at home and housebreaking to maintain their loan recovery record. Such observations indicate that women's empowerment through financial inclusion should be examined by investigating gender roles and relations in the existing gender power structure [60,61].

## Gender socialisation and performance in families

A central concern of this paper is the patriarchal gender relations between spouses and the gendered norms related to decision-making about microfinance loans. In societies where patriarchal norms are core to the social power hierarchy, men dominate women in decision-making process. However, both men and women (re)produces gender relations and ascribe discriminatory gender norms to maintain women's typically inferior position in households [38]. Social institutions such as marriage [62], the religious and social practice of domestic confinement or seclusion of women, and gender division of labour (e.g., household chores and childcare responsibilities), economic dependency on men, low-wage practices, and formal political power imbalances [63] allow men to sustain control and appropriate the fruits of women's labour [36].

Another social and religious institution of women's seclusion is *purdah*, a religious doctrine for females to cover their bodies and not to meet or greet unknown people [64]. Particularly, in *purdah*, "the gender division of labour is grounded in values of honour (*izzat*) and modesty or shame (*lojja*) expressed in the ideal of female seclusion" [64]. Cain, Khanam [65] found that patriarchal constraints such as *purdah* system put on women to lessens their mobility and economic opportunities. They are kept inside within household boundaries undertaking tasks such as food preparation and processing, household maintenance, animal husbandry and child care, while men work beyond the household domain on largescale cultivation, trading, extensive farming and transportation which has more potential for economic gain [65].

Patriarchy is institutionally sustained with the reproduction of inter-generational gender norms and gender relations, and may also induce new gender norms over time to (re)capture men's supremacy over women [66]. Therefore, as a system that produces discriminatory male-female power relationships, patriarchy can be addressed by shifting norms and practices inter-generationally in families, in communities or in larger societies.

In this paper, we use gender socialisation and gender performance theory to examine these everyday micro-level or household-level gender relations and the reproduction of patriarchy in Bangladesh. Gender socialisation theory assumes that gender identity relevant to gender roles is a culturally driven process, learned and internalised by individuals in the community.

Gender identity is formed within social structures and interaction; gender identities are viewed as a product of regular social interactions [41]. Goffman [39], West and Zimmerman [40], Butler [37] theorise gender performance theory in line with gender socialisation theory and argue that by performing gender roles in everyday interactions, long-lasting gender norms are enacted, produced, reproduced and sustained within social structures [41]. In a patriarchal setting, gender socialisation processes and gender performance over time provide an ideological legitimacy for men's role and make them take hierarchical positions over women according to the hegemonic model [41,67].

Furthermore, family, parents, peer groups, partners, social institutions such as schools and religious bodies are important agents of gender learning and socialisation [41]. Among these agents, family, parents and partners are the most powerful for individuals because many of the gender interaction takes place at the household level and are highly influenced by family members throughout their life [41]. John, Stoebenau [41] explain that,

> . . .these interactions are repeated over and over again in everyday life, individuals learn the gender differences in expectations, values, preferences and skills, and adapt their own behaviour accordingly to ultimately form their gender identity in line with the prevailing gender norms in their social environment.

Family members (mainly parents) influence young children and teach them how to behave at the household level, particularly how to perform their expected roles in the family such as the division of labour and gender stereotyped activities to carry out expected gender behaviours and relations [Leaper and Farkas cited in 41].

Many scholars [36,64,65,68–71], among others, investigate male-dominated gendered power structures, gender norms and practices in Bangladesh. However, these studies do not examine how household relations and the reproduction of gender norms may be affected by microfinance schemes. The evidence presented in this section shows how women and men learn, perform and (re)produce gender norms in Bangladesh and if this is affected by financial factors and constitutes a new approach generating new knowledge in the field of microfinance. The gender socialisation process can either be further entrenched by microfinance loan management and decision-making processes. The processes of change advocated for cannot be assumed with changes to gender roles and socialization practices.

## Methodology

This study collected primary data in southern *Khulna*, Bangladesh from locations with at least 15 years of microfinance history. The *Khulna* district was purposively selected because of the large eligible sample size within its sub-districts and 364 operating microfinance institutions [72]. The *Dumuria* sub-district was randomly selected from the nine [9] sub-districts of Khulna district. A mixed-method design was used to collect quantitative (survey) and qualitative (in-depth interview) data from our study site. Data were collected from August 2018 to May 2019. This study received approval from the Monash University human research ethics board.

This study focuses mainly on women's understanding and experiences of microfinance in a southern sub-district (*Dumuria*) of rural Bangladesh. However, husbands of microfinance recipient women were also consulted to understand the male perspective on gender and microfinance. The criteria for identifying informants in this study included: i) they had to be current female microfinance borrowers; ii) they should have at least two living children (one male and one female) and iii) the women have or had at least one surviving brother. The

criteria are chosen to discover patterns of gender socialisation and gender performance as discussed in the theoretical discussion above. We assume that a mother who had grown up with a brother(s) and have at least a son (or more) and a daughter (or more), would provide more comparative, valid and reliable data on household gender norms and its practices than a mother who had no brother or having either son(s) or daughter(s).

The present study comprised two stages of data collection. In the first stage, a representative sample of 331 female recipients of microfinance was surveyed using simple random sampling technique (confidence interval of 4.93 at 95 percent confidence level) from a population of 2048. It is to be noted that the research team found no existing population list (based on criteria for selection as informants) to use and therefore had to manually create the list by using a Researcher Administered Informal Census (RAIC) including a set of questions to ascertain eligible population of this study. The RAIC list (who agreed to take part in our data collection process under condition of securing confidentiality of their identity) identified 2048 eligible informants for our data collection. In the second stage, approximately 33 in-depth interviews were undertaken with microfinance beneficiaries (26 microfinance recipients and seven recipients' husbands) who had taken part in the survey and consented that they would also be interviewed.

A semi-structured interview schedule in *Bengali* was prepared for data collection. The survey included demographic questions relating to household structure, the microfinance scheme being accessed, and household gender norms being observed. Before conducting the field survey, 15 interview schedules were tested to minimise inconsistency and maximise validity and reliability of the instrument. In-depth interviews were scheduled after the surveys took place at locations chosen for the informant's comfort and convenience and to maximise the likelihood of garnering frank and detailed responses. An open-ended and loosely-structured question format was developed, with topics ranging from microfinance participation; household gender practices; microfinance influencing or negotiating household gender practices from recipients and their husbands, and; participants' experience of inter-generational gender norms. Informed written consent was secured prior to conducting the survey and interviews after providing participants with an explanatory statement. The survey took around 30 to 40 minutes and the in-depth interviews both took 45 to 60 minutes on average. The survey data has been processed and analysed in SPSS for descriptive statistics. The in-depth interviews were transcribed in Bengali and then translated into English. Interview data were coded and thematically analysed and combined with background survey data to arrive at the ultimate findings.

## Limitations of the study

This study is conducted on a purposively selected district (*Khulna*) in Bangladesh. Therefore, the study location (*Dumuria* sub-district) does not provide a full representative picture of the whole country. Moreover, the selection of the microfinance receiving households also have a clear criterion, therefore, the respondent women of this current research does not represent the wider group of microfinances borrowing women in rural Bangladesh. However, the findings of the current study may be potentially important while comparing results with different microfinance recipient groups in different locations.

## Findings/Results

This section presents the quantitative data from our field survey on the overview of women's loan pattern and qualitative data outlining gendered roles and relations regarding finance.

## Overview of loan patterns and gendered decision-making

Survey findings indicate that approximately 71% of the study sample are first-generation borrowers and 27% are second-generation microfinance borrowers. Of the 331 women surveyed, most were aged between 30–39 (48.9%) and 40–49 (25.7%), with most (44.1%) having undertaken formal education up to the primary (Class I-V) level and 32% attended secondary (Class VI to X) level. Among the women, most (41.1%) of them started taking loan between 2001 to 2010, 19.6% started between 1988 to 2000 and the rest starts after 2010. Regarding loan frequency during their microfinance tenure, 31% of women reported taking 3 to 5 loans 42% took 6–10 loans; and 27% of women took more than 10 loans.

Quantitative data further indicates that the length of participation in microfinance has not resulted in any significant changes in women's control over finance. Although the microfinance loans were formally given to our surveyed women, in most cases the money was used and controlled by men (e.g., husband, son). Over 60% reported (see Fig 1 their husbands made the decision to take out their loans from the first, to the most recent loan. In terms of loan use (see Fig 2), most of the women reported that men undertook the investment activities (e.g., fish farming, business and purchasing a van) as the primary purpose of the loan. However, a significant amount of loans is also used for social reproduction purposes (e.g., household consumption, repairing house, daughter's marriage), or repaying other loan or debt. Approximately 80% (see Fig 3) of the loan was also repaid by borrower's husbands.

Qualitative findings explained the logic behind the behaviour: 25 of the 33 interview participants, including microfinance recipients and their husbands, agreed with the convention that men must conduct financial dealings and therefore must take control over of the microfinance loan. The majority (17) of the female informants were found to have handed over the loan money to their husbands (e.g., for uses including the construction of *gher* [traditional ponds], petty business, buying a vehicle or house repair), and at least five informants handed over loan money to their sons for uses ranging from medical treatment, education and business use. One thirty-six-year-old interviewee, Beauty (all the names used for informants in this research are

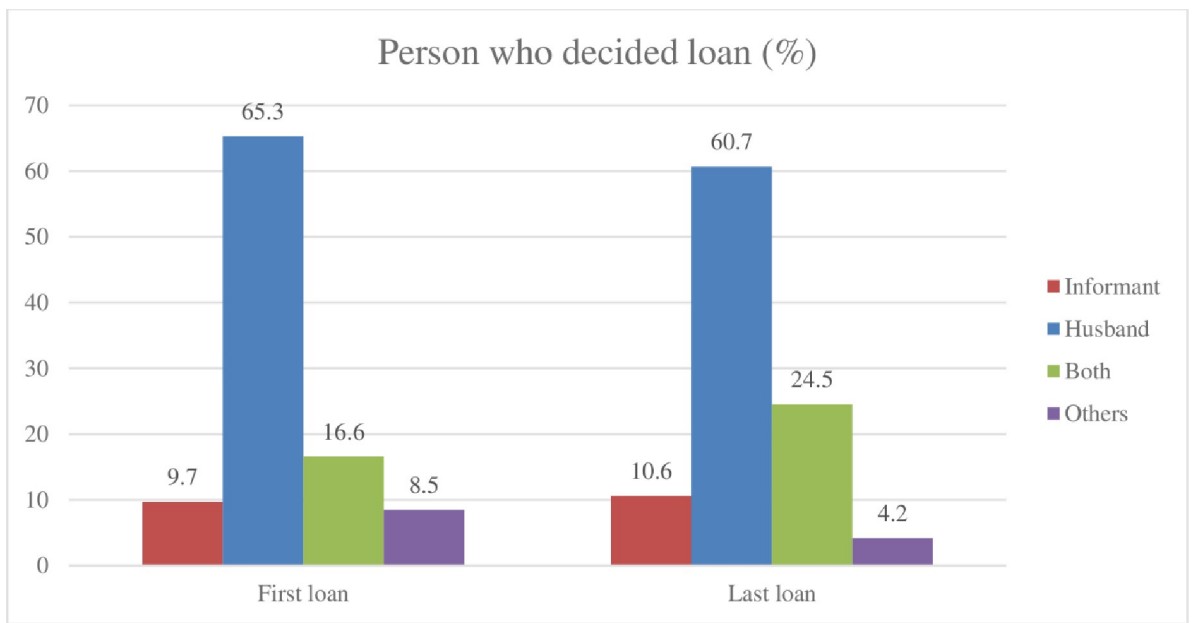

**Fig 1. Loan taking decisions.**

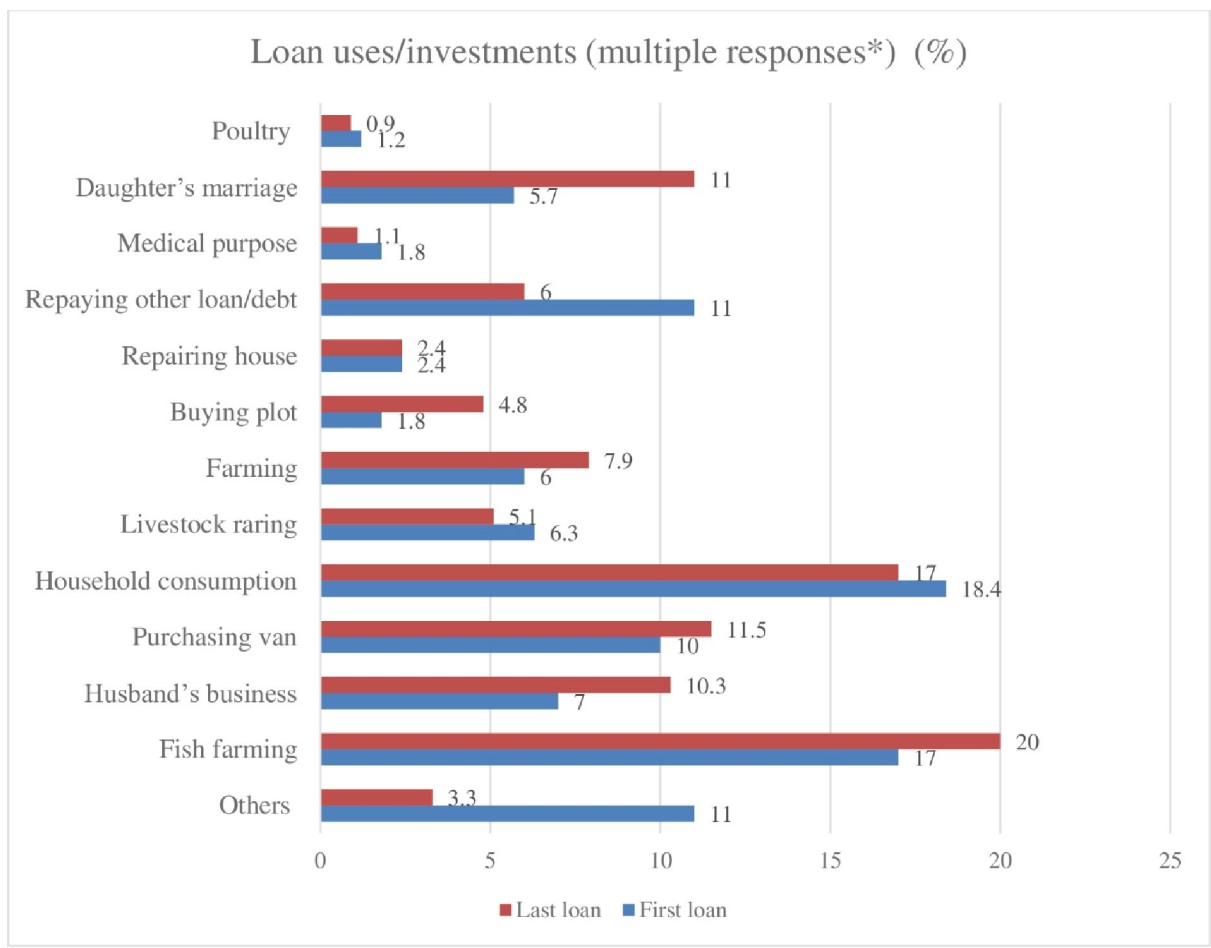

**Fig 2. Use/investment areas of the microfinance loan.**

pseudonym, considering the safety, security and privacy concern of them), a second-generation microfinance borrower, explained the practice this way: "I do not use the microfinance loan. I borrow the money for my husband's use. He works outside and invests the money in his business. Men are good investors".

Only two informants described using a share of their loan money in partnership with their husbands and son. Many (17) of the informants argued that microfinance requires regular repayment, however, traditionally women have no income-generating opportunities with which to make repayments because they are engaged in full-time domestic responsibilities. Many (16) of the informants, both recipients and their husbands, stated that they abide by/or believe in traditions that dictate that only men can engage in paid productive labour and additionally should control family finances. Seven of the female informants also agreed that women's restrictions from paid labour left them without any income or access to money, meaning that only men as income-earners with access to money could repay the loan instalment. This means in practice that women hand over their loan money and its management to men. Informants consistently asserted that men being responsible for managing family finances is a practice that has existed for generations. In the sections below, we describe the underlying gender norms, socialisation practices and beliefs that help to reproduce and reinforce this practice.

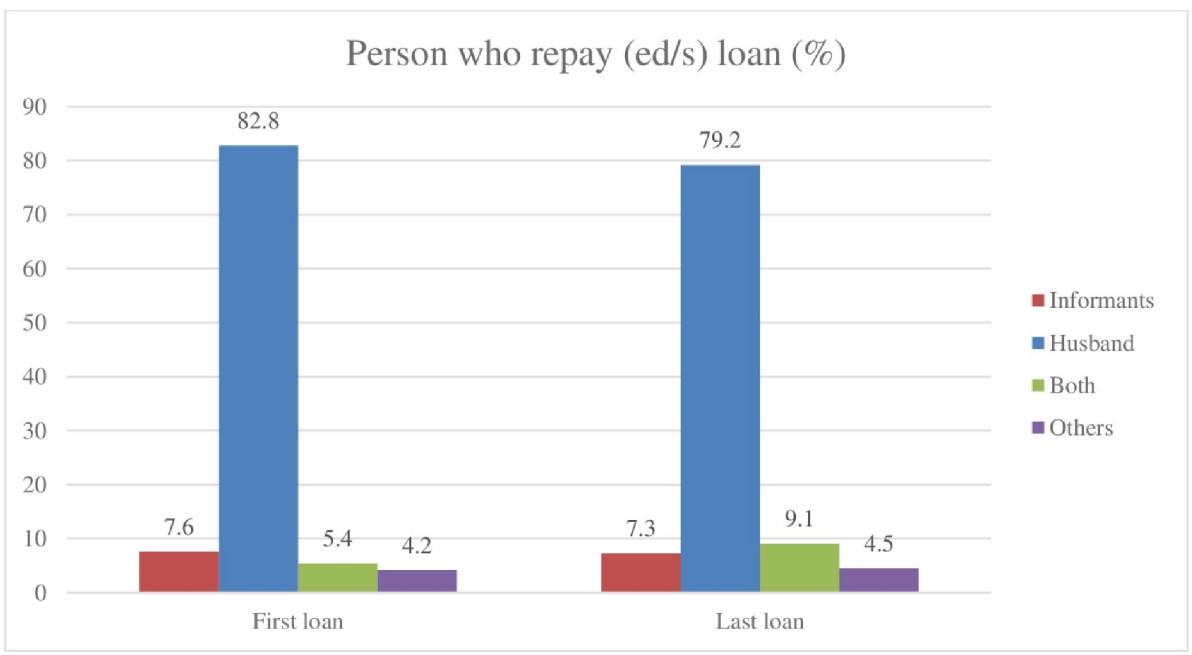

**Fig 3. Loan repayments.**

## Reciprocally mutual duties: Inter-generational reproduction of the gendered division of labour

Our survey data (see Figs 4 and 5) found that women mainly engaged in unpaid domestic labour (95%), including serving/preparing food, cleaning dishes, washing clothes, cleaning

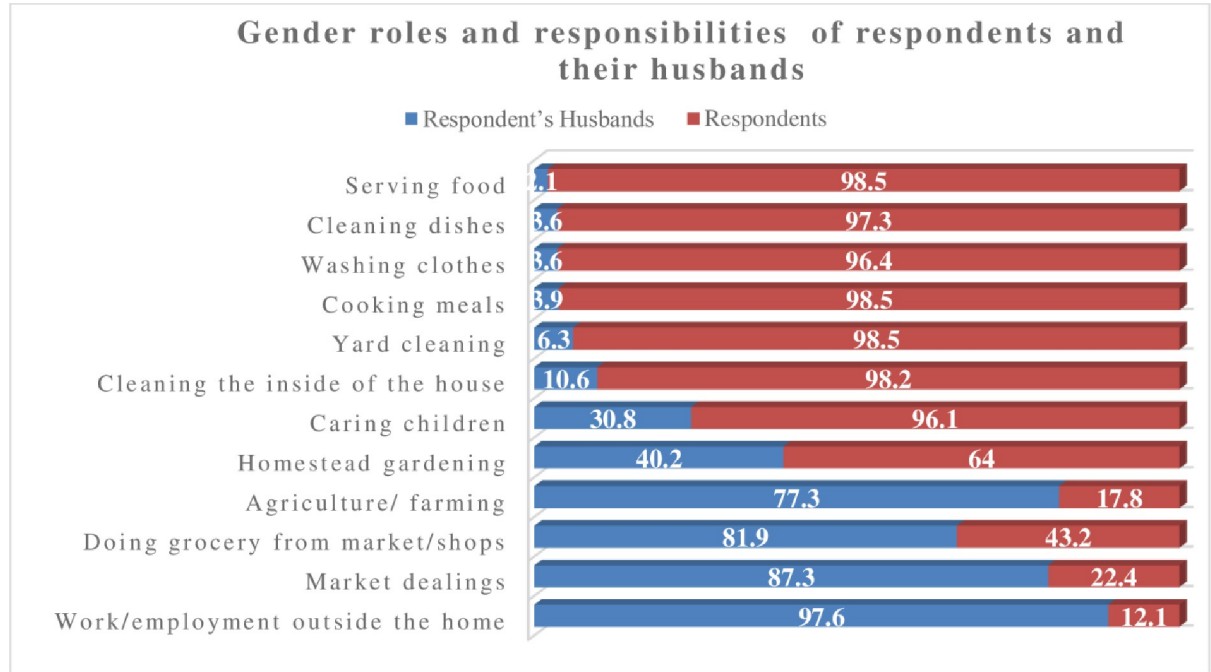

**Fig 4. Household gender role and responsibilities of the respondents and their husbands.**

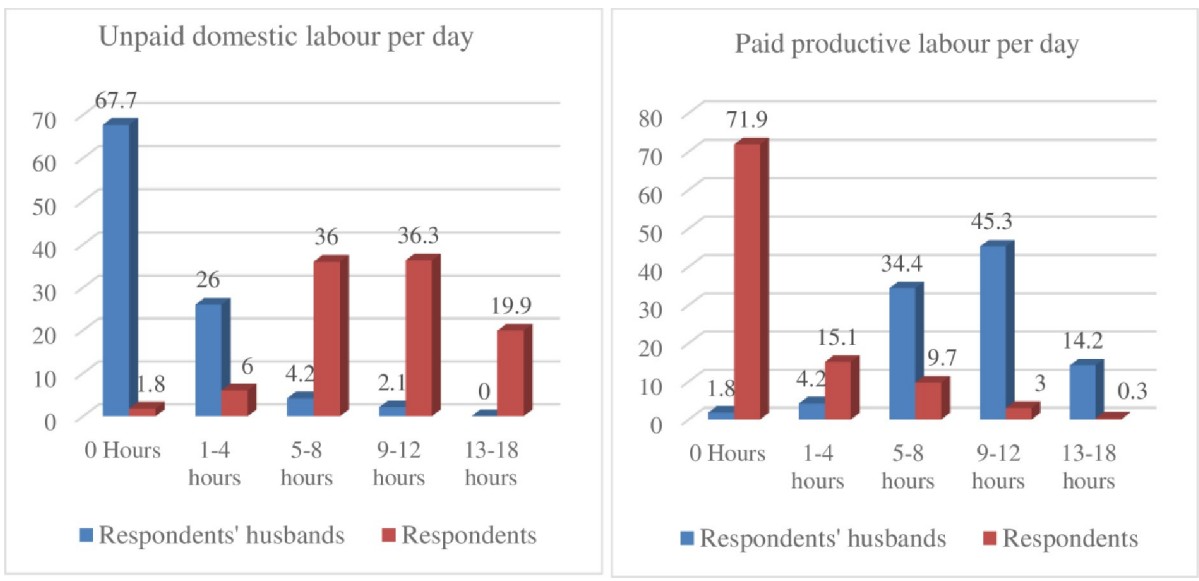

**Fig 5. Paid/unpaid labour hour spend by respondents and their husbands.**

yard and inside of home and caring for children. Approximately 57% of women surveyed undertook unpaid domestic labour for 9 or more hours per day, and the remaining 42% for up to 8 hours per day. Almost all (95%) stated that their mother, grandmother, sister also carried out such domestic responsibilities. Conversely, almost 80% of husbands were involved in paid productive labour such as agriculture, market activities or other employment outside the home. More than 80% of the women surveyed also said their fathers, brothers, grandfathers were also involved in paid labour.

In-depth interviews uncovered why women took responsibility for hours of unpaid domestic labour and men were assigned to caring for the role of an earner. Twenty-five (25) interviewees, both microfinance recipients and their husbands, agreed that it was the wife's responsibility to manage unpaid domestic labour and that husbands were responsible for paid productive labour. They indicated that they have been socialised with these gender norm and roles for men and women since childhood and that they are continuing this practice with their children.

Twenty-two (22) of the interviewees, both microfinance recipients and their husbands, referred to the gender role of their parents or grandparents, whom they remembered complying with a similar gendered division of labour. They described these gender roles and relations as social convention, which they viewed as inevitable and which must be followed and similarly carried out by the next generation(s). For example, thirty-nine-year-old interviewee, Morjina, a second-generation female borrower commented on carrying out the inter-generational tradition, "all day I labour in the home *(ghorer kaj. Ghorer kaj*, this is a Bengali term that is used by the informants to refer all the unpaid domestic labour [e.g., cooking, cleaning, collecting water, preparing meals, feeding children, watering homestead garden, taking care of small-scale household livestock (e.g., poultry, cattle)] women provide at their household. *Ghor* means home/household and *kaj* means labour/labour related responsibility. By tradition, women were assigned these responsibilities from generations.*)* to support my family. I remember my parents and grandparents also carrying out such tradition. I do the same, so will my daughter".

Identifying their domestic socialisation as accepted tradition, 22 female informants also remembered that they helped their mother in her unpaid domestic labour. One forty-three-year-old interviewee, Safiya, a first-generation female borrower, stated:

I remember my mother worked at home her entire life, serving as an unpaid domestic labour *(ghorer kaj)*, which we were taught was a women's responsibility *(meyeder kaj. Meyeder kaj*, locally this term is used by the informants to refer Bengali women's responsibility of managing unpaid domestic labour at home. *Meye* means girl. As women take the responsibility of unpaid domestic labour, these are labeled under their responsibility list.*)*. This includes all the domestic work for our family, such as household cleaning (e.g., washing dishes and laundry), cooking and preparing meals.

The same 22 female informants remembered that men were occupied with income-generating activities outside the home, and that their brothers worked with their fathers', as Safiya illustrated further,

My father's work was performed outside the home *(bahirer kaj. Bahirer kaj*, this Bengali term informants used for men to refer their responsibility of paid productive labour (e.g., day labour, agriculture, driving, construction work, fishing, van pulling, trading) outside the home. *Bahir* in Bengali means outside the household boundary. By *bahirer kaj*, men were identified with the responsibility of income-generation from engaging paid productive labour. Most of the paid productive labour took place outside the home, and therefore, they call it as *bahirer kaj*. It also used alternatively with *cheleder kaj. Chele* means men, *kaj* means labour. As men we assigned with *bahirer kaj*, it also possesses synonymous meaning for *cheleder kaj.*); his focus was to undertake the roles understood as men's responsibility *(cheleder kaj. Cheleder kaj*, locally this term is used by the informants to refer Bengali men's responsibility of managing paid productive labour outside the home. *Chele* means boy. As men take the income-generation responsibility, paid work is labeled under their responsibility list.*)*. My father was a farmer, and he mainly worked in the field. He sold his agricultural products at the market and brought the money home for the family. I never remember my father or brothers being asked to provide a hand toward my mother's responsibilities *(meyeder kaj)*. Mostly, my sisters and I helped my mother doing her domestic work.

Many (18) of our study informants believed that the traditional gendered division of labour was inevitable and considered this as 'reciprocally mutual duties'. To informants this meant a set of normative gender roles and relations, traditionally women carrying out domestic labour and men taking responsibility for income-generation, on which the harmony of the family relies. For example, one thirty-five-year-old interviewee- Lipika, a second-generation borrower stated:

Men's traditional responsibility is to earn money and provide for the family's needs such as food, clothing, health, housing. My husband maintains his responsibility *(bahirer kaj)* by involving in paid productive labour, and this income feeds us. In return, my tradition dictates that I carry out my domestic responsibilities *(meyeder kaj)*. If I manage my unpaid domestic labour *(ghorer kaj)* well, this is an excellent support to my husband's endeavours for our family.

Most (18) informants said that this continued observance of strict gender roles serves as the foundation for men's total control over family finances. A comment made by Aklima, a forty-

five-year-old first-generation female borrower, "men deal with the money, and therefore, men must earn the income" is indicative of this consistently-held view. Four female interviewees reported that doing the opposite is considered as a violation of the norm. Aklima's comment on women violating their traditional roles reflects how strong community beliefs are about these practices:

> . . .some women go outside the home and undertake daily paid labour *(bahirer kaj)*. They are disobeying our tradition, and their husbands. If they spend hours in paid labour, then who carries out their domestic responsibilities? We observe them having quarrel *(jhogra. Jhogra is a Bengali term that refers to quarrel between individuals or among people. It denotes verbal aggression showed to one another. However, mostly informants referred one-sided verbal aggression they experience from their male counterpart when their husbands are angry with or dissatisfied of any issues.)* with husband, and we accuse them of bringing unhappiness *(oshanti. Oshanti is another Bengali word that means turmoil/trouble. Informants argued that if their husbands are dissatisfied with them, the situation creates lots of tension for their family. No woman wants to dissatisfy their husbands in any issue to ensure harmony in their family. Therefore, they do not want to go against their husband's decisions. Informant argued that community practice also encourages women to manage family happiness (shanti) by any means. Shanti means peace/tranquility, which is the opposite of oshanti.)* to their family. I feel pity for them. Women must remember their role and duties. Their priority should be their domestic labour. Undoubtedly, working outside the home is men's duty.

All (n = 7) the informants' husbands also supported the above perception. They also concluded that their hard work and labour is a sacrifice for their family and that women are required to conduct unpaid domestic labour in obligation for men's income generation, as indicated by Islam Zoddar, a sixty-year-old husband of a first-generation borrower:

> If my wife participates in paid productive labour, then who will manage household responsibilities? Who will cook, clean my house and manage other household duties? Men earn income for their family, and the wife must manage household duties. This is the formula for family happiness. This is what we traditionally see and learn from previous generations.

Many (18) women also referred to their experiences back to their early life in their parents' house. One thirty-six-year-old interviewee, Beauty, a second-generation female borrower said:

> . . .my husband bears all the cost of my subsistence such as clothes and food. He never asked me to pay for that. In return, I focus on managing my domestic responsibilities. I perceive that leaving my domestic responsibilities and working outside the home in paid productive labour would belittle my husband's sacrifice. During my childhood at my parents' house, I remember my father and mother carrying out similar roles and responsibilities as my husband and I do in our own home.

Majority (25) of the informants' experiences showed that conventional gender norms were learned and practised within the household whereby men took responsibility for income-generation the family income and women took responsibility for domestic duties as part of traditionally prescribed reciprocal gender relations. Nevertheless, the reproduction of a gendered division of labour is not solely responsible for the restriction of women to unpaid domestic labour. In the section below, we explore the importance of *Purdah* on female microfinance recipients.

**Purdah.**   During our in-depth interviews, it emerged that a further impediment for women's financial empowerment was *purdah*. As mentioned in the introduction P*urdah* is a set of social exclusion practices grounded in religious doctrine that regulate the seclusion of women in the home, enforce their exclusion from public spaces and establish specific gender identities [64,65,73,74]. *Purdah* can include decrees that a woman cover her face and a major portion of her body, not meet or greet unknown people, and even not appear in common space such as marketplaces, public thoroughfares or playgrounds. *Purdah* is "centrally about subjugation of women and is sustained by potent cultural and religious system, the net result of which is that observance of *purdah* grants status and prestige while non-observance removes status" [74]. Early studies [64,65,74] conducted in Bangladesh found that women's *purdah* and norm regulated behaviour had limited their economic participation in the community. However, Hossain and Kabir [75], Agarwal [76] argued that one-way *purdah* norm has been challenged by rural women is by microfinance participation. They found that due to microfinance participation the recipient women challenge *purdah*, move outside the home and take part in IGAs.

*Purdah* was understood by many of our interviewees (17 interviewees both recipients and their husbands) as a social norm which must be maintained by women themselves to ensure that there would not be dishonour or shame bought to her or her family. They agreed that women's engagement in paid productive labour violates *purdah*. Halima, a forty-three-year-old first-generation female borrower expressed this sentiment:

> . . .going outside and working with men will violate our *purdah*. Our tradition teaches us to maintain our *purdah* first. Since my childhood, my parents taught me to comply with the rules of *purdah*. I also taught my daughter to observe *purdah* carefully. In our community, women are taught this must be maintained. Therefore, if men earn the family income for their family as they should, I believe, women must not be involved in paid employment and violating *purdah*.

A few (six) female informants argued that engaging in paid productive labour requires mobility and communication with previously unknown people, both of which violate *purdah*. One forty-eight-year-old interviewee, Nur Jahan, a first-generation borrower asserted that:

> . . .the commercial employment sector is not suitable for women. A woman needs to work with other men and must work like a man. While working, she may not be able to cover her body with clothes, all the time. A woman communicating or jointly working with other men outsider the family is a violation of *Shariah law. Shariah law is r*eligious (Islamic) doctrine for a Muslim that explains how to carry out everyday life and interaction process for the satisfaction of *Allah* and Muslim community.. It also disrupts her *purdah* observance.

Six interviewees also argued that maintaining *purdah* provided women with a greater sense of security. They understood that *purdah* is a preventive measure to safeguard their honour. Limiting mobility and not being involved in paid labour outside the home protected a woman's exposure to potential harassers. Instead of holding the harasser accountable, the *purdah* norms assert that it is the woman that needs to safeguard herself from potential harassment. Nur Jahan further commented:

> . . .a woman's non-observance of *purdah* might provoke sexual harassment. A woman must protect her *purdah* and honour (*izzat*. *Izzat* is a Bengali terminology used frequently for women's honour in the community. Scholars such as White [64] also used this term referring to women's honour in the community. *Izzat* is an important cultural phenomenon for

a woman to uphold and preserve her respect in the community. Study informants also argued that women's honour requires to maintain cultural norms and perceptions assigned to them. Such as maintaining *purdah*, following *Sariah law*, avoiding public spaces (e.g., marketplaces, playground) and men's (considerably outsiders) communication, avoiding situations like harassments, not involve in a romantic relationship before marriage, not allowing people any reason to spread gossip. Furthermore, women are prescribed to staying within home boundary and being obedient to father or husband. *Izzat* is a perception about women which is both physical and abstract. Physical perception denotes women to preserve their honour from being harassed or sexually abused by any man or avoid any consensual intimate relationship before marriage. The abstract perception referred to community perception about a woman's character and personality. If a woman is found not following community norms and regulations assigned for women, she loses her honour (*beizzat*) in the eye of the community. *Beizzat* is antonym of Bengali word *izzat*.). And the best way to protect her honour is by staying at home. Therefore, since childhood, we were discouraged to work outside the home. This is sad but true that we hear some news about women's harassment by men at a workplace or on their way to work. In both places, a woman does not have her male family members (e.g., father, brother, husband) to protect her from the harasser.

Most (17) of our interviewees (both recipients and recipient's husbands) further commented that *purdah* as a traditional gendered practice that restricts the mobility of women and girls and subsequently impacts their ability to trade in the market. They argued that the market is men's space, and traditionally, women do not go because it violates women's *purdah*. Six informants indicated that community censure results if women violate *purdah* and visit the marketplace. One forty-four-year-old interviewee, Nasima, a first-generation female borrower said that "trading in the market would be disastrous. People may spread negative gossip about me and defame *(beizzat)* myself and my family". She further commented on attending the market to trade or purchasing goods:

> We, women, are taught to avoid visiting the market because of *purdah*. Women are traditionally discouraged from meeting and greeting other men in public space, such as a marketplace. We may go there occasionally to buy clothes or cosmetics; however, we must maintain proper clothing for *purdah*. In terms of trading, I remember my father and uncle (s) doing that, and now my husband and brother(s) do it".

At least eight women argued that they never consider participating in trading in the market the products they make from their micro-finance enterprise. The three women who reported running small-scale household-based businesses (e.g., poultry, livestock, sewing) relied on male relatives to trade their produce. Parvin, a forty-year-old first-generation female borrower, whose husband had abandoned her, reported that:

> I have a small poultry business and a cow borrowed from BRAC. Sometimes, I need to sell milk, chicken and eggs. My husband left me, and being a woman, I must not visit the marketplace. If I visit there to trade my goods, people will question if I am observing *purdah*, stigmatise me and, repudiate my honour, (*izzat*). Therefore, when I need to sell my goods, I ask my brother for help. He takes the product to the market, negotiates the price, and sells it for me.

As mentioned early, majority (18) of our study informant argued that oftentimes, women's participation in microfinance was limited to only taking out the loan because of gendered

division of labour. The gender roles and norms described above by our participants allow men to take control over use, investment and repayment of microfinance loans. As the quote above demonstrates even if women produce goods at home they typically rely on male relatives for pricing, marketing or selling produce so as to safeguard their honour in the eye of the community.

## Discussion

Patriarchal gender norms in Bangladesh dictate that income-generation is men's responsibility, therefore the women in our study surrendered control of their money, including microfinance loans, to male relatives. This finding concurs with findings in the literature about the cultural practice of money management being attributed to men that result in microfinance borrower women relinquishing control of loans to their male family members (e.g., husbands, sons) [61,77–79]. Additionally, this study validates the findings that women were used as a channel to access microfinance money for men Chin [78], Kabeer [79], Ali [80] and Goetz and Gupta [27]. The current study contradicts findings of an earlier body of literature arguing that by providing financial access to women microfinance participation helps them to engage in IGAs [1,4,19,81]. Furthermore, this study contributes a deeper understanding to how these strict gender roles and norms that result in women's lack of financial power are practiced and re-entrenched within families which remain unaffected by participation in microfinance programs.

We found that among the microfinance recipient households, social convention and traditional practices about the gendered division of labour ensure men's role in paid productive labour outside the home and women's roles as unpaid domestic labour within the home. Labour participation allows men not just mobility and freedom of movement, and conversely women's restriction to the home, but is also linked to income-generation and controlling the proceeds of productive labour making monetary dealings men's responsibility. The restriction on women's movement formalised in the practice of *purdah* not only shapes women's expected gender role to be confined and labouring unpaid in the home but reduces options for women to earn an income from paid labour or any claim or control of the proceeds. This self-perpetuating cultural system framed as a reciprocally supportive arrangement between husband and wife requires deconstruction. Most importantly, the study informants, both microfinance recipients and recipient's husbands have demonstrated that they were Socialised into these strict gender roles, practices and norms from early childhood. That is these gendered, behaviours are learned cultural practices within families and households. Participants also reported reproducing these gender roles, practices and norms within their own families and inculcating their children with them, thereby ensuring their perpetuation across generations.

A key element of the patriarchal social practices experienced by the women in our study was *purdah*, which restricted recipient women's mobility and reduced women's opportunity to earn an income or be involved in paid productive labour such as microfinance-enterprise trading. The current study identified that *purdah* significantly restricted women's mobility and subsequent ability to engage in IGAs. Our participants made it clear that practicing *purdah* was crucial to upholding their own and their families honour *(izzat)* and creating the community's perception of them. They described how a woman also may be sexually harassed if she discontinues her *purdah* maintenance and appears in a public place, pathways or in a workplace. Therefore, for safety, security and protecting women's honour *(izzat)*, families dictate and strictly monitor women's *purdah*.

Microfinance beneficiaries (including husbands) reported replicating *purdah* with their children making *purdah* a practice that continues to define women's lives and roles restricted

to the household. As a result, female microfinance beneficiaries are often unable to change or challenge their unpaid domestic roles or to participate in microfinance enterprises that would require violating their *purdah*. Few of the women participants (in these cases always either a divorcee or widow), who had a small-scale enterprise of their own, were found to seek the assistance of men while conducting trading in the marketplace to maintain *purdah*.

The current study challenges some of the main findings of scholarly literature investigating microfinance in rural Bangladesh, such as Mahbub [82], Pitt, Khandker [83], Schuler, Hashemi [84] and Sharma [85], who state that women's microfinance participation improves their mobility and their participation in the market. It may be women's microfinance participation has improved their mobility and their participation in the market in some other areas but not in southern Khulna. The current study corroborates findings of some the scholars such as Mahmud, Shah [57], Greeley [68] and Davis [86] who focused on patriarchal gender norms in rural Bangladesh to evaluate women's restricted mobility. These scholars showed that in rural Bangladesh, gender norms governing women's mobility is strictly observed. They conducted studies with non-microfinance recipients, however this current study contributes to this literature by adding the experiences and analysis of microfinance recipient women's restricted mobility. This is significant because it shows the enduring nature and imperviousness of these social practices in the face of interventions that aim to shift them. Importantly it also shows the lack of impact of microfinance interventions on patriarchal beliefs and practices in rural Bangladesh.

Findings from this study show that patriarchy as a system is reproduced through gender socialisation processes of male-domination, of particular importance in our study are those relating to financial matters. Nevertheless, the strict gender division of labour and mobility-restricting practice for women such as *purdah* prevent them from not only participating in IGAs but also having any agency in financial dealings such as investing and repaying money, contributing or spending money on family needs, family budgeting, trading and market dealings. Last but not the least, although the scope of this article does not cover education as discussion point, but we see that our recipient women are less educated. No doubt that interventions in the education sector for women would help them overcome the gender and purdah norms/practices. This study findings leaves scope for future researchers to investigate the role of education and relevant interventions to challenging purdah restrictions in rural Bangladesh.

Therefore, women's financial empowerment will require a more expansive intervention than microfinance participation, such as changing everyday gender norms and practices, recognition of women's unpaid labour, improving women's education: providing access to money alone cannot financially empower women. Participation in microfinance programs does not challenge the foundations of patriarchy which keep women subordinate to men and disempowered. The persistence of patriarchy and the deep socialisation to its values and practices that we found in the microfinance receiving households in our study requires a much deeper reconceptualization and renegotiation of gender power relations and gender roles than microfinance programs provide in women are to be empowered, financially or otherwise. Unchanging practices such as *purdah*, and gender norms relating to the gendered division of labour (including money management) and gender socialisation limit the transformative and empowering potential of microfinance participation by women. Our findings therefore reinforce earlier research [57] that demonstrates how important understanding gender norms and socialisation practices are to appreciating the impact of microfinance participation on beneficiaries.

## Conclusion

This study aimed to examine the impact of gender norms on the financial empowerment of women recipients of microfinance in rural Bangladesh. We found that patriarchal norms

dominate everyday life in rural Bangladesh, and that participating in microfinance programs cannot ensure women's financial empowerment at the household level. Instead, it is the dominant patriarchal household norms that dictate financial participation for women rather than the intentions or interventions of MFI loans. This research demonstrates that patriarchal gender norms enforce men's control over women's loans and that assumptions about women's financial empowerment processes as a result of microfinance program participation need to be re-evaluated. Our findings reinforce the finding that majority of the microfinance loans are decided, controlled, used and repaid by men. The strict gendered division of labour and its inter-generational reproduction explained most clearly in theories of gender socialisation and gender performance theory reinforced restrictions on women and entrench men's dominance in economic matters. Microfinance beneficiaries (both women and their husbands) have identified that reproduction of inter-generational gender-related norms and practices reinforce a set of gender values and roles for men and women at odds with the aims of microfinance programs. Men's roles were to perform classic 'breadwinner' activities, while women are assigned domestic duties with little financial power. Moreover, the norm of *purdah* for women also constitutes a restriction on their mobility throughout their lifetime. As a result, women receiving microfinance very rarely experienced any sense of financial empowerment during their years of participation in microfinance schemes.

Though the study made every effort through a probability sampling (simple random sampling technique) to understand the wider group of microfinance recipients of rural Bangladesh, their gender dynamics at household level, and participation in microfinance programs, we acknowledge that this research only provides the understanding and experiences of the microfinance recipients of this study location; their perception cannot be generalised to the wider population of Bangladesh, which would require a larger sample conducted all over the country. Although this study did not intend to provide a fully representative picture of the perceptions of different microfinance recipient groups of women in the nation of Bangladesh, the responses and experiences explained here are so consistent that they may be important for comparing results of how different microfinance recipient groups experience these important issues.

Overall, of this study finds little evidence that providing financial access to women ensures their financial empowerment, instead it reveals that financial empowerment is not just determined by access to finance, but by focussing on the everyday gender norms and practices that men and women learn, nurture and reproduce throughout their lives. Substantive financial empowerment for women, as for other forms of empowerment, will depend on the transformation of these gendered power relations. This study makes an original contribution to the literature on microfinance which the industry and further research may consider while evaluating or approaching women's financial empowerment process. Ultimately, the findings of this study force us to rethink prescriptive linear articulations of women's financial inclusion and empowerment.

## Author Contributions

**Conceptualization:** Tunvir Ahamed Shohel.

**Data curation:** Tunvir Ahamed Shohel.

**Formal analysis:** Tunvir Ahamed Shohel.

**Investigation:** Tunvir Ahamed Shohel.

**Methodology:** Tunvir Ahamed Shohel.

**Supervision:** Sara Niner, Samanthi Gunawardana.

**Writing – original draft:** Tunvir Ahamed Shohel.

**Writing – review & editing:** Tunvir Ahamed Shohel, Sara Niner, Samanthi Gunawardana.

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
