## [Decision Letter · Decision Letter 0]

26 Oct 2020

PONE-D-20-29072

How the persistence of patriarchy undermines the financial empowerment of women microfinance borrowers in rural Bangladesh

PLOS ONE

Dear Dr. Shohel,

Thank you for submitting your manuscript to PLOS ONE. After careful consideration, we feel that it has merit but does not fully meet PLOS ONE’s publication criteria as it currently stands. Therefore, we invite you to submit a revised version of the manuscript that addresses the points raised during the review process.

The study is very interesting but needs some adjustments in title, methodology, results and conclusions sections. For details, please see my comments below this letter. I also suggest to adequately address reviewers' comments.

We look forward to receiving your revised manuscript.

Kind regards,

Abid Hussain

Academic Editor

PLOS ONE

Journal Requirements:

Additional Editor Comments (if provided):

Dear Authors,

It is an interesting research. However, needs revisions in different sections. Please see below my concerns.

Study district was selected possessively. And selection of respondent women was also based on a clear criteria. These tow steps clearly show that study is non-representative for rural areas of Bangladesh. Even it s not representative for all microfinance borrowing women in study district. But the title is framed and the way results interpreted, it is claimed to be a representative study. I suggest to present this study as an evidence (with mixed methods) of a specific group of rural borrowers from Khulna district, Bangladesh. There is no harm at all in presenting this study as an evidence.

To address above suggestion, I suggest to adjustments in title, results interpretation and implications in conclusions section. Also, clearly mention in the methodology about the limitations of the study, i.e. non-representative, and have weak external validity (can not be generalized for rural Bangladesh, even for rural borrowing women of Khulna Distt.).

I hope these suggestions will help improving the quality of the manuscript. Fro further comments, please reviewers comments.

Reviewers' comments:

Reviewer's Responses to Questions

**Comments to the Author**

1. Is the manuscript technically sound, and do the data support the conclusions?

Reviewer #1: Yes

Reviewer #2: Yes

2. Has the statistical analysis been performed appropriately and rigorously? 

Reviewer #1: N/A

Reviewer #2: No

3. Have the authors made all data underlying the findings in their manuscript fully available?

Reviewer #1: No

Reviewer #2: Yes

4. Is the manuscript presented in an intelligible fashion and written in standard English?

Reviewer #1: Yes

Reviewer #2: No

5. Review Comments to the Author

Reviewer #1: 1. The citations format allows to mention the author name and respective governments are considered the author of all the official reports; rather mentioning the name of particular organization. For example “Microfinance Regulatory Authority, 2018 at Pg-6”.

2. The specified characteristics defined for the baseline to identify the informants of this study may lead to create a biases as well as rigid sample and in the presence of such sample it becomes difficult to generalize the results for the entire women borrowing community.

3. Because of the fact that the study based upon the primary information and thus it is imperative to employ suitable statistical techniques i.e. Cronbach's Alpha, to examine the reliability of data.

4. In the presence of such a rich information, the use of regression analysis may be a good addition and also more interesting; rather using the simple percentages. Hence, it is suggested to use an appropriate econometric model i.e. Logit or Probit, to empirically determine the facts & figure for the policy derivation.

5. No doubt the study is very interesting, but it lacks the suggested remedial measures to overcome this issue.

Reviewer #2: “How the persistence of patriarchy undermines the financial empowerment of women micro-finance borrowers in rural Bangladesh”- This is a very interesting and important study. The results and discussions are very rich with insightful information. The findings from this study, findings from the rural Bangladesh is very alarming and will provide important basis for policy and development interventions. However, the study needs some improvements here and there to improve its overall quality.

My constructive comments are as below.

Introduction

Page 2 Microfinance has been widely recognised as an ideal type of aid and development intervention for enhancing women’s financial empowerment by increasing women’s income, financial contribution to household and financial decisionmaking power in the family (Ackerly, 1995; R. Amin & Becker, 1998; Assassi, 2009; Duflo, 2012; Grameen Bank, 2011; Khandker, 1998).

- Have any relevant publications appeared more recently? It would be worth a quick look and give latest references as well.

- Recommended proofing to help to correct minor errors.

In this paper, we define patriarchy as “a system of power in which male privilege and

superiority over women are manifest, institutionalised, and self-reproducing across a society as a whole” (Shepherd, 2019, p. 116).

In this paper, we define patriarchy as “a system of power in which male privilege and

superiority over women are manifested, institutionalised, and self-reproduced across a society as a whole”

- The introduction part is nicely woven. It mentions “This mixed-method, multidisciplinary study examines the impact of gender norms on the financial empowerment of women recipients of microfinance in rural Bangladesh.” However, it doesn’t explicitly explain what is the objective of the study. In the same paragraph authors better add what is expected from this study, expected study implications.

- This methodology and result sections are not necessary in Intro section.

To explore the issues above, we conducted 331 surveys and 33 in-depth interviews with women receiving microfinance and their husbands in the Dumuria sub-district in the southern region of Bangladesh. Bangladesh is globally recognised as the home of microfinance since the mid- 1970s (Yunus & Jolis, 1998). We found that women’s financial empowerment is significantly undermined by entrenched gender norms and customary gender practices, regulated by an ontology of patriarchy. We argue that it is important to consider how women receiving microfinance and men internalise gender norms about finance and financial participation and that this is important to understand their financial empowerment outcomes.

Gender socialisation and performance in families

- What is Purdah? It should be explained when used for the first time (first use) in the manuscript. In the current sentence “Social institutions such as marriage (Delphy, 1984), purdah (White, 1992),…..” it is unclear. Please rearrange or rewrite the sentence.

This may present a gap in current knowledge, since the existing evidence presented in this section of the paper indicates that how women and men learn, perform and (re)produce gender norms in Bangladesh is influenced by financial factors; a woman receiving microfinance assistance and her spouse might be performing crucial roles about gender they have learned through the gender socialisation process, which may both influence and be strengthened by microfinance loan management and decision-making.

- Present or May present? If authors are presenting the research/knowledge gap, they should be sure about that, not “may present a gap”. Long sentences like this one should be avoided. It is hard for readers to follow. Such long sentences need rechecking/revision throughout the manuscript.

Methodology

It is about how you collected data. So, it should be “this study collected primary data in southern Khulna,……” It’s not “this study presents….” in the methodology. It needs some language twists.

This study focuses mainly on women’s perceptions and experiences of microfinance in Bangladesh.

- As I understood, the study is performed in Dumuria sub-district of southern Khulna city. Does the study in this particular area represent the whole Bangladesh? I think the situation between city and rural Bangladesh must be different too. Can the result be generalized for whole Bangladesh? If not, then it should be revised accordingly.

- The baseline for identifying informants of this study specified some characteristics: i) they had to be current female microfinance borrowers; ii) they should have at least two living children (one male child and one female child) up to 15 or more years old, and; iii) the women have or had at least one surviving brother (at least 15 years of old) while growing up in the family home.

The baseline (better to say criteria)… This sentence needs rewriting to improve the flow. Thorough editing is essential help to improve the quality, readability of the paper.

We assume that a mother who had grown up with a brother(s), and have at least a son (or more) and a daughter (or more), would provide more comparative, valid and reliable data on household gender norms and its practices than a mother who had no brother or having either son(s) or daughter(s).

- I think both gives valid and reliable data but it’s the matter of different perspective that you are looking for from the earlier one for comparative analysis.

- Is there any reason why only 10% of the survey sample was chosen for in-depth interviews?

- Is the semi-structured interview guide/schedule included along with this paper? That will be helpful for the readers to understand what kind of questionnaire is used for particular data collection. How long did the survey and in-depth interview took in average for study participants?

- Methodology lacks details on data analysis. How the collected data are analyzed? E.g. Simple statistics, and what else?? Is it possible to do any statistical analysis???

Findings/Results

Overview of loan patterns and gendered decision making

It will be effective to present the quantitative data/results in bar diagrams than in words.

Purdah

However, Hossain and Kabir (2001) Agarwal (1997) argued that one-way purdah norm has been challenged by rural women is by microfinance participation.

- One-way purdah norm has been challenged by rural women is by microfinance participation. I agree but in overall, I think educating women (their access and continuity to (school and higher) education) is essential to change the purdah norm and control of male-counterparts in microfinance participation/decision-making. How education may affect gendered decision making, inter-generational reproduction of the gendered division of labour in future needs to be discussed.

Discussion

The current study challenges some of the main findings of scholarly literature investigating microfinance in Bangladesh, such as Mahbub (2001), M. M. Pitt, Khandker, and Cartwright (2003), S. R. Schuler, Hashemi, & Riley (1997) and Sharma (2008), who state that women’s microfinance participation improves their mobility and their participation in the market.

- It may be- women’s microfinance participation has improved their mobility and their participation in the market in some other areas but not in southern Khulna. How about this?

These scholars showed that in Bangladesh, gender norms governing women’s mobility is strictly observed.

- Is it throughout the country or only in rural areas?

Importantly it also shows the lack of impact of microfinance interventions on patriarchal beliefs and practices in Bangladesh.

- Is this just because of purdah. I think it is largely because of no interventions in the education sector for women. If education is improved then it will improve the purdah norms and others consequently. This should be discussed if it’s within the scope of this study.

Therefore, women’s financial empowerment will require a more expansive intervention than microfinance participation: providing access to money alone cannot financially empower women.

- Again, I think it is largely because of no interventions in the education sector for women. If women are educated then they will overcome the gender and purdah norms/practices. Education will automatically empower women and their decision-making power consequently. This should be discussed if it’s within the scope of the study.

Conclusion

Although this study did not intend to provide a fully representative picture of the perceptions of different microfinance recipient groups of women in the nation of Bangladesh, the responses and experiences explained here are so consistent that they may be important for comparing results of how different microfinance recipient groups experience these important issues.

- Authors must revise the earlier section in the similar way to reflect that it is not a full representative picture of the whole country. As of now, in earlier sections, authors talk about whole Bangladesh. So, it is confusing. See my comments above.

6. PLOS authors have the option to publish the peer review history of their article (what does this mean?). If published, this will include your full peer review and any attached files.

Reviewer #1: No

Reviewer #2: No

---

## [Author Response · Author response to Decision Letter 0]

15 Mar 2021

Response to reviewers has been uploaded with the revised file.

---

## [Editor Report · Decision Letter 1]

30 Mar 2021

How the persistence of patriarchy undermines the financial empowerment of women microfinance borrowers? Evidence from a southern sub-district of Bangladesh

PONE-D-20-29072R1

Dear Dr. Shohel,

We’re pleased to inform you that your manuscript has been judged scientifically suitable for publication and will be formally accepted for publication once it meets all outstanding technical requirements.

Kind regards,

Abid Hussain

Academic Editor

PLOS ONE

Additional Editor Comments (optional):

Dear authors,

Thank you for addressing the concerns raised by me and reviewers adequately.
---

## [Editor Report · Acceptance letter]

5 Apr 2021

PONE-D-20-29072R1 

How the persistence of patriarchy undermines the financial empowerment of women microfinance borrowers? Evidence from a southern sub-district of Bangladesh 

Dear Dr. Shohel:

I'm pleased to inform you that your manuscript has been deemed suitable for publication in PLOS ONE. Congratulations! Your manuscript is now with our production department. 

Kind regards, 

on behalf of

Dr. Abid Hussain 

Academic Editor

PLOS ONE